# Meta-Analysis on Energy-Use Patterns of Cropping Systems in Iran

**Morteza Zangeneh** [1,*] , **Narges Banaeian** [1,*] and **Sean Clark** [2]

1 Department of Agricultural Mechanization Engineering, Faculty of Agricultural Sciences, University of Guilan, Rasht 4199613776, Iran
2 Department of Agriculture and Natural Resources, Berea College, Berea, KY 40404, USA; clarks@berea.edu
* Correspondence: zanganeh@guilan.ac.ir (M.Z.); banaeian@guilan.ac.ir (N.B.);
Tel.: +98-918-9110115 (M.Z.); +98-918-0195190 (N.B.)

**Abstract:** We present a meta-analysis of energy-consumption and environmental-emissions patterns in Iranian cropping systems using data collected from articles published between 2008 and 2018 for 21 different crops. The results show that the crops consuming the most energy per hectare are tomato, sugarcane, cucumber and alfalfa, while sunflower consumed the least. The average total energy input for all crops in Iran during the study period was 48,029 MJ ha$^{-1}$. Our analysis revealed that potato has the highest potential to reduce energy consumption and that electricity and fertilizer inputs have the most potential for energy savings in cropping systems. Not all studies reviewed addressed the factors that create energy consumption patterns and environmental emissions. Therefore, eight indicators were modeled in this meta-analysis, which include Total Energy Input, Energy Productivity, Energy Use Efficiency, Net Energy, Greenhouse Gas Emissions, Technical Efficiency, Pure Technical Efficiency and Scale Efficiency. The effects of region (which was analyzed in terms of climate), year and crop or product type on these eight indicators were modeled using meta-regression and the nonparametric Kruskal–Wallis test. To create a comprehensive picture and roadmap for future research, the process of the agricultural-systems analysis cycle is discussed. This review and meta-analysis can be used as a guide to provide useful information to researchers working on the energy dynamics of agricultural systems, especially in Iran, and in making their choices of crop types and regions in need of study.

**Keywords:** energy indicators; environmental assessment; greenhouse gas emissions; cropping systems; climate; sustainability

## 1. Introduction

In recent years, the number of publications on energy and environmental aspects of agricultural systems has increased in Iran, particularly on energy consumption patterns and environmental emissions [1]. The considerable number of publications in this field of research has necessitated a comprehensive and analytical study of the various dimensions of such studies. The method of conducting studies in this field has evolved in recent years. However, there are challenges facing researchers in this field. The first question of the present study is: Despite the wide range of indicators and methods used by previous researchers, have all aspects of cropping systems been examined or are there still issues in the systems analysis process that have been neglected? Given the great variety of production systems in the agricultural sector, have the necessary standards been met by researchers in sampling and data collection? In terms of the system analysis process, it is incomplete to study the current state of a system without intervening to improve a system. Therefore, the next question is whether, in the studies, interventions in the energy consumption pattern have been done by researchers and whether the effect of those interventions on improving the energy consumption pattern, economic model and reducing environmental emissions has been studied? Another important question is whether a hypothesis has been made about the reason for the formation of energy consumption

patterns in the products under study. Are solutions proposed to improve the current state of a system in the studies? Have social indicators been studied in studies in this field? Has the selection of the region and the product under study been based on regional and national needs and necessities? In response to the above questions, and in order to provide better directions for future research and prevent duplicate studies that have sometimes been observed in this area, it is necessary to provide researchers with sufficient information about the background of the studies. Therefore, using a systematic method, the present article provides readers with a comprehensive picture of the research background and energy consumption status in Iranian cropping systems. The sections that have been neglected in previous research are also mentioned in this article, which can be used as a basis for future research by researchers to improve the quality of studies in this field. A new procedure to supplement the shortcomings of previous studies is introduced in this article. Moreover, understanding the energy-use patterns of cropping systems is essential to addressing the sustainability challenges of agriculture. Thus, a systematic review of the relevant literature is necessary for assessing what is known and establishing future research agendas, particularly where the results of published empirical studies have generated apparently conflicting findings. A preliminary review of these studies shows that in some cases, research on the same product in different parts of the country has generated notably different results. For example, the energy efficiency index of a barley crop in Hamadan province was reported to be 2.86 [2], while in Isfahan province, it was 1.43 [3]. The existence of such differences is the rationale for this meta-analysis, which is aimed at interpreting, understanding and generalizing wherever possible based on studies published from 2008 to 2018.

Meta-analysis can be used to combine evidence from different studies into a single statistical framework, explain differences arising from conflicting findings, and identify necessary areas for future research. Iran's relatively large geographic area and climatic diversity results in a range of practices and systems for crop cultivation and leads to differences in reported values for energy and environmental indicators. The objective of study is to provide an overview of current understanding and analyze the impacts of crop type, region and year on energy and environmental indicators of cropping systems of Iran.

## 2. Methods

### 2.1. Review Protocol

This study follows PRISMA guidelines and is a companion article to one published recently on trends in methodologies and future research directions in crop energy analyses [1]. In this study, a meta-analysis was performed using published studies of energy-use patterns in cropping systems of Iran. We independently selected papers, extracted data, and assessed the results using the review process illustrated in Figure 1. Essentially, we began with the question of why the results of energy and environmental indicators for the same products differed between similar studies.

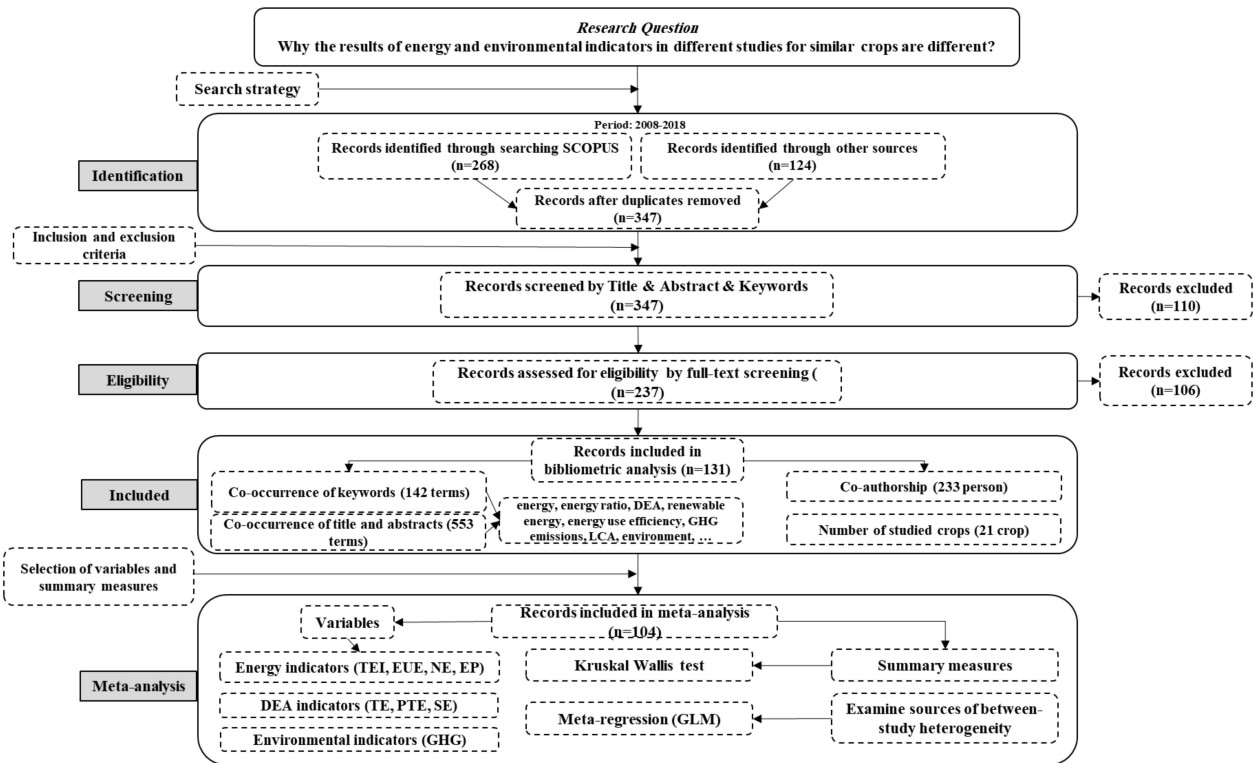

**Figure 1.** Schematic structure of the systematic review process.

### 2.2. Article Selection

The selection process included several steps described in detail in [1]. Potentially relevant titles were identified with keyword searches on SCOPUS and Google Scholar. The searches were restricted to crop studies performed in Iran, and all papers from reputable journals were given consideration. Then, abstracts and keywords were reviewed, and some inappropriate papers omitted.

### 2.3. Data Collection

To collect data needed for the meta-analysis, the full texts of articles were carefully reviewed, and the published results were recorded and categorized by crop or product type, study location, and year of research. Then, the values for indicators related to energy consumption patterns were extracted from the articles, including total energy input (TEI), energy use efficiency (EUE), energy productivity (EP) and net energy (NE). The type and share of each input as an amount of energy consumed were recorded. The highest t-ratio and highest marginal physical productivity (MPP) were collected as well. This initial data-extraction step was applied to all the articles reviewed.

The second step applied only to studies of cropping systems using Data Envelopment Analysis (DEA). The methodology of DEA and its application in energy analysis of cropping systems is described well by Banaeian et al. [4]. In this part, total energy savings (TES), technical efficiency (TE), pure technical efficiency (PTE) and scale efficiency (SE) were collected, as well as the shares of all inputs in the energy-saving target ratio (ESTR). The third step pertained to studies analyzing greenhouse gas (GHG) emissions and included the quantity of GHG emitted in form of $CO_2$ equivalents (both land-based and mass-based). In cases with more than one study of a product, the average value of an indicator was determined. For example, there were eight studies found on watermelon energy consumption [5–12], so the reported indicator values are derived from the average from all these papers.

Energy and GHG Conversion Factors

Indices used in this meta-analysis were calculated using two series of coefficients (Tables 1 and 2). Energy equivalents (energy conversion factors) are numerical values that reflect the state of energy for each input. These equivalents are used to estimate the energy indicators using collected sample data. When the amount of a consumed input is multiplied by the conversion factor, the result is equal to the energy of the input. There are some highly relevant inputs common in agricultural systems, but the energy system boundary for each is not the same in all energy studies. Table 1 shows the most relevant inputs and energy equivalents applied in energy-use-pattern studies of Iranian agricultural systems.

**Table 1.** Main inputs and energy equivalents used in agricultural energy systems.

| Inputs (Unit) | | Energy Equivalent (MJ per Unit) | References |
|---|---|---|---|
| Labor (h) | | 1.96 | [13] |
| Machinery (h) | | 62.7 | [14] |
| Diesel fuel (L) | | 56.3 | [15] |
| Fertilizers (kg) | Nitrogen | 66.14 | [16] |
| | Phosphate | 12.44 | [16] |
| | Potassium | 11.15 | [3] |
| | Micro | 120 | [3] |
| Farmyard Manure (kg) | | 0.3 | [17] |
| Chemicals (kg) | Herbicide | 356.29 | [18] |
| | Pesticide | 280.44 | [18] |
| | Fungicide | 181.9 | [18] |
| | Insecticide | 101.9 | [19] |
| Water (m$^3$) | | 1.02 | [20] |
| Electricity (kWh) | | 11.93 | [21] |

The $CO_2$-eq emissions coefficients shown in Table 2 were used to calculate the amounts of the GHG emissions from crop-production inputs in Iran. The application rates of machinery, diesel fuel, farmyard manure, electricity, chemical fertilizers and biocides per hectare or per unit of product were multiplied by their corresponding emissions coefficients from Table 2.

**Table 2.** Greenhouse gas (GHG) emission coefficient of inputs [22].

| Inputs (Unit) | | GHG Coefficient (kg $CO_2$-eq Unit$^{-1}$) |
|---|---|---|
| Machinery (MJ) | | 0.071 |
| Diesel fuel (L) | | 2.760 |
| Fertilizers (kg) | Nitrogen (N) | 1.300 |
| | Phosphorus ($P_2O_5$) | 0.200 |
| | Potassium ($K_2O$) | 0.200 |
| Farmyard Manure (kg) | | 0.126 |
| Biocide (kg) | Herbicide | 6.300 |
| | Fungicide | 5.100 |
| | Insecticide | 3.900 |
| Electricity (kWh) | | 0.780 |

*2.4. Risk of Bias in Individual Studies*

A risk in studies like this one is the inability to generalize the selected sample to the entire statistical population. Researchers have often used Cochran and Neyman methods to determine the number of statistical samples. However, an important issue that researchers should consider when sampling agricultural systems is the diversity of these systems. It is preferable to select samples from all types of agricultural systems in a given area. Thus, it is necessary to first introduce the characteristics of agricultural systems in the study area and perform sampling operations based on the variety of existing systems. Failure to provide

enough information about agricultural systems and non-compliance with the proportion between the number of samples and the statistical population poses a risk to the accuracy and precision of the data as well as the values of the indicators calculated in such studies. In this regard, steps to reduce such risk in the individual studies included in this review are not possible.

### 2.5. Summary Measures and Synthesis

The results of this article are summarized in two sections: bibliometric analysis and meta-analysis. In the bibliometric section, the co-occurrence of keywords and co-authorship was examined using the title, abstract and keywords of the studied articles and visualized using VOSviewer software [23]. Several assumptions were made: (1) the keywords used in these texts were carefully selected by the authors; (2) the use of different keywords within a text requires some relationship between those words and; (3) the co-occurrence of keywords in the texts by different authors means that the relationships between these words are important. The meta-analysis section identifies patterns in the results of the studies included, sources of disagreement among those results, or other interesting relationships. Reasons for observed heterogeneity among the results of similar studies were interpreted using the nonparametric Kruskal–Wallis test and meta-regression. After collecting the necessary data, the results were summarized graphically. Then, for the meta-analysis, the data were analyzed using three factors: crop (product), year and regional climate.

Iran's total geographic area is 1,648,195 km$^2$, and the diversity of climates has made it possible to cultivate a variety of agricultural products across different provinces. To account for the influence of climate on cropping systems, the country was divided into 12 climate regions based on [24] (Figure 2) and each study was assigned to the appropriate climate. Table 3 lists the main climates and provinces allocated to them. It was possible for an individual province to be found within more than one climate. In such cases, the climate that occupied the largest area of a province was considered the main climate of that province and served as the basis for further analyses. Regression modeling was performed to examine the effects of crop, year and climate in each of the indicators. The relationships between each of these three factors and each of the eight indicators, which included EUE, TEI, NE, EP, GHG, TE, PTE and SE (see abbreviations), were analyzed individually using the Kruskal–Wallis test.

**Table 3.** Classification of Iran's provinces into different climates.

| Climate | Provinces |
| --- | --- |
| Cold Semi Dry | East Azerbaijan, West Azerbaijan, Zanjan, Qazvin, Alborz |
| Very Wet and Mild | Ardabil, Guilan |
| Wet and Mild | Mazandaran, Golestan |
| Semi Dry Hot | North Khorasan, Razavi Khorasan, South Khorasan |
| Extreme Desert and Very Hot | Sistan and Baluchistan, Kerman |
| Semi Warm Desert | Isfahan, Semnan, Yazd |
| Very Hot Coastal Desert | Hormozgan |
| Warm Semi Mountains | Fars |
| Warm Coastal Desert | Bushehr, Khuzestan |
| Cold Mountains | Kohgiluyeh and Buyerahmad, Chaharmahal and Bakhtiati, Markazi, Lorestan, Kurdistan |
| Moderately Semi Wet | Kermanshah, Hamedan, Ilam |
| Temperate Desert | Qom, Tehran |

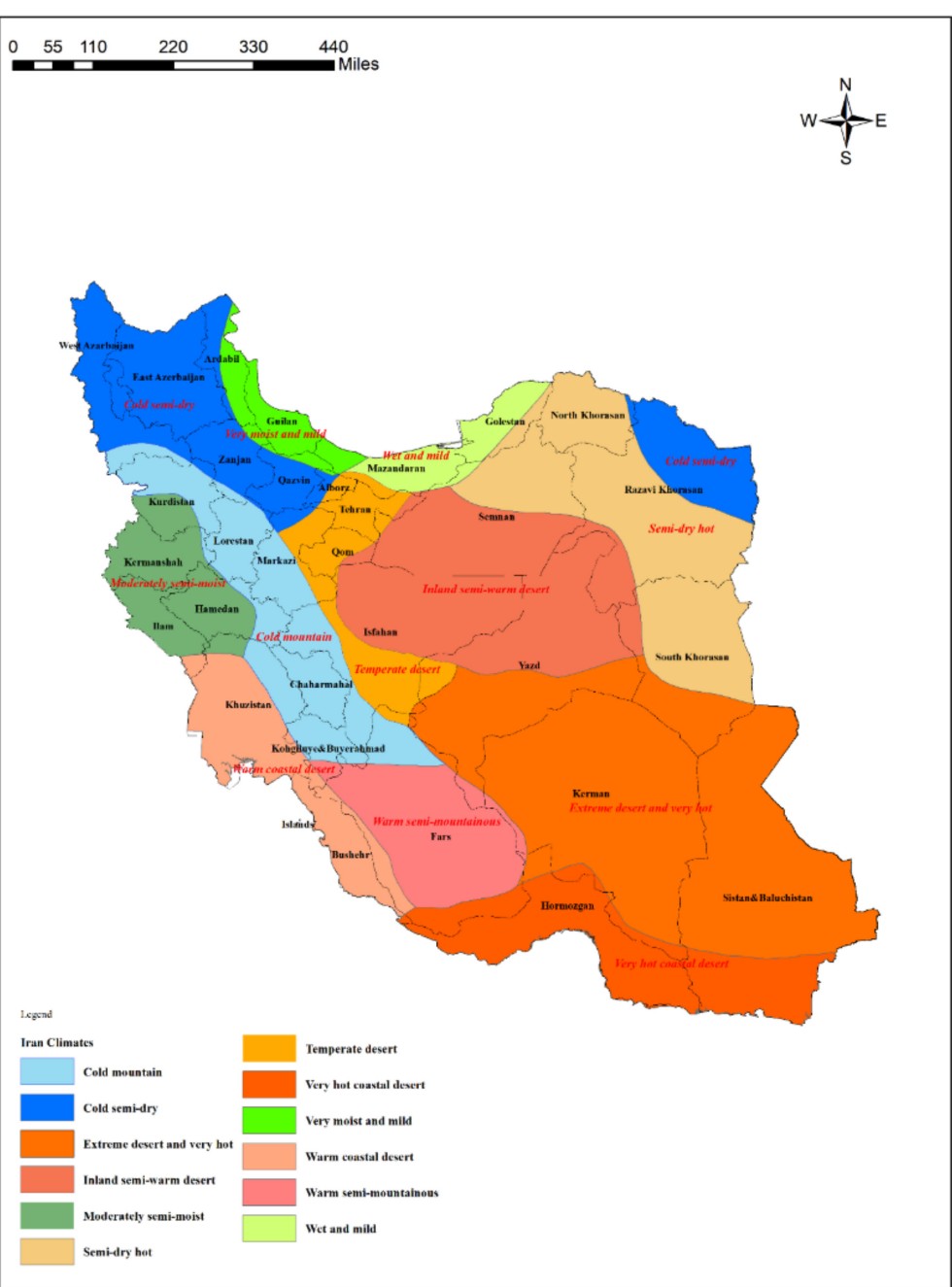

**Figure 2.** Climates and provinces of Iran [24].

### 2.6. Risk of Bias across Studies

Typically, when collecting data on input consumption in crop studies, other conditions influencing the production system are assumed to be constant. However, factors such as soil conditions, climate, available technology, knowledge of producers, land-use systems, social factors, input prices, market prices, market freedom status and the extent of government intervention in market regulations (especially in Iran), among other factors, affect input consumption as well as crop production. Lack of attention to these factors has resulted in large differences in indicator values reported in different studies. The intent of this review is to interpret the heterogeneity of results in such studies based on available data.

Statistical Analysis

Regression using the General Linear Model (GLM) was performed to identify variables responsible for variations in all dependent variables, namely energy and environmental indicators. GLM usually refers to conventional linear regression models for a continuous response variable given continuous and/or categorical predictors. To determine whether the association between the response and each term in the model was statistically significant, the *p*-value for the term is compared to the significance level to assess the null hypothesis. The null hypothesis is that there is no association between the term and the response. Due to the nominal nature of the variables related to the cropping systems (year, climate and crop), the non-parametric Kruskal–Wallis test was used as an alternative to one-way ANOVA. Statistical significance at 0.05 and 0.01 levels was determined.

## 3. Results

### 3.1. Article Selection

A total of 131 articles were ultimately selected and used for bibliometric analysis and 104 of these were included in meta-analysis. The journals with the most published articles analyzing energy consumption patterns in Iranian agricultural systems were *Energy*, *Journal of Cleaner Production*, *Renewable and Sustainable Energy Reviews*, and *Energy Reports*, which each contained more than four articles from 2008–2018.

### 3.2. Study Characteristics

Quantitative and graphical summaries of word co-occurrences in the keywords, titles and abstracts among the 131 articles using VOSViewer software are shown in Table 4 and Figure 3, respectively. A threshold limit was used to plot the co-occurrence network. The minimum number of occurrences for keywords was 4 times. Accordingly, 15 keywords were found at least 4 times in articles to be related to other keywords used by researchers. For the terms used in the titles and abstracts, the threshold was set at 5 times. There were 25 terms found to meet this condition. Out of 233 authors who participated in 131 reviewed articles, 18 of them contributed more than 5 times to articles. How these terms relate to each other is shown in Figure 3.

The following terms were the most common in the articles: energy, greenhouse gas emissions, data envelopment analysis, sensitivity analysis, economic analysis, energy efficiency and technical efficiency. Shahin Rafiee, a faculty member at the University of Tehran, has the most joint publications, co-authoring 43 papers between 2008 and 2018. There were 18 authors found to have published more than 5 articles.

**Table 4.** Bibliometric analysis of the selected articles.

| Description | Categories of Co-Occurrences | | |
| :---: | :---: | :---: | :---: |
| | Keywords | Titles and Abstracts | Authors |
| Total items | 142 | 553 | 233 |
| Minimum number of occurrences | 4 | 5 | 5 |
| Number of items meet the threshold | 15 | 25 | 18 |

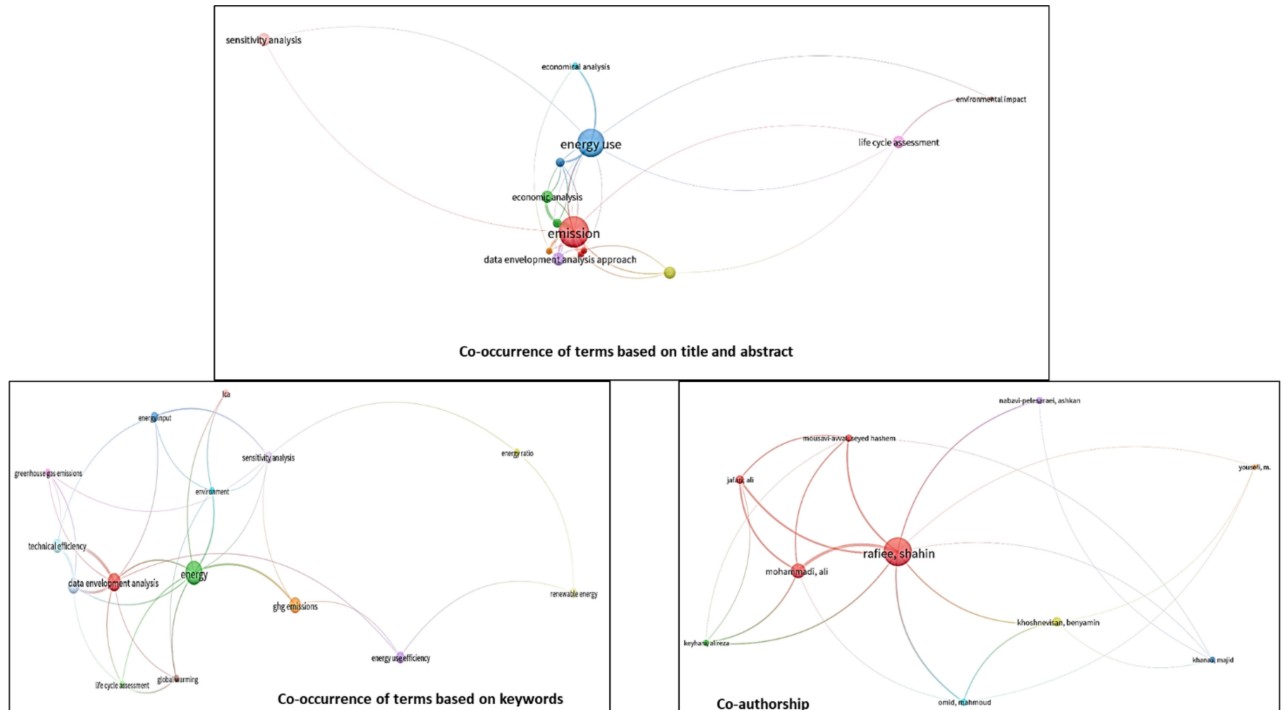

**Figure 3.** Graphical display of the co-occurrence analysis.

### 3.3. Synthesis of Results

To create a useful overview of energy consumption in Iran's cropping systems, we first present a summary of the most important results published on input consumption, environmental impacts, and data envelopment analysis. Then, the effects of the independent variables of year, climate and crop on the dependent variables of energy and environmental indicators are analyzed and modeled. Finally, the relationship between each of the independent and dependent variables is analyzed using the non-parametric Kruskal–Wallis test.

#### 3.3.1. Share of Inputs

A percentage breakdown of the TEI into input categories for each crop is illustrated in Figure 4. For crops in which there was more than one article, averages are used. Fertilizer and fuel are the main consumers of energy in many crop systems. Authors often report the share of important energy-consuming inputs and make recommendations for improvements to reduce the TEI. For example, Hosseinzadeh-Bandbafha et al. [25] recommended that to benefit from a balanced use of fertilizers, emphasis should be placed on using high-quality seeds, optimal timing, precisely timed irrigations and better agronomic practices in farm operations (e.g., reduce field passes and using shallow tillage) for peanut farms at the north of Iran.

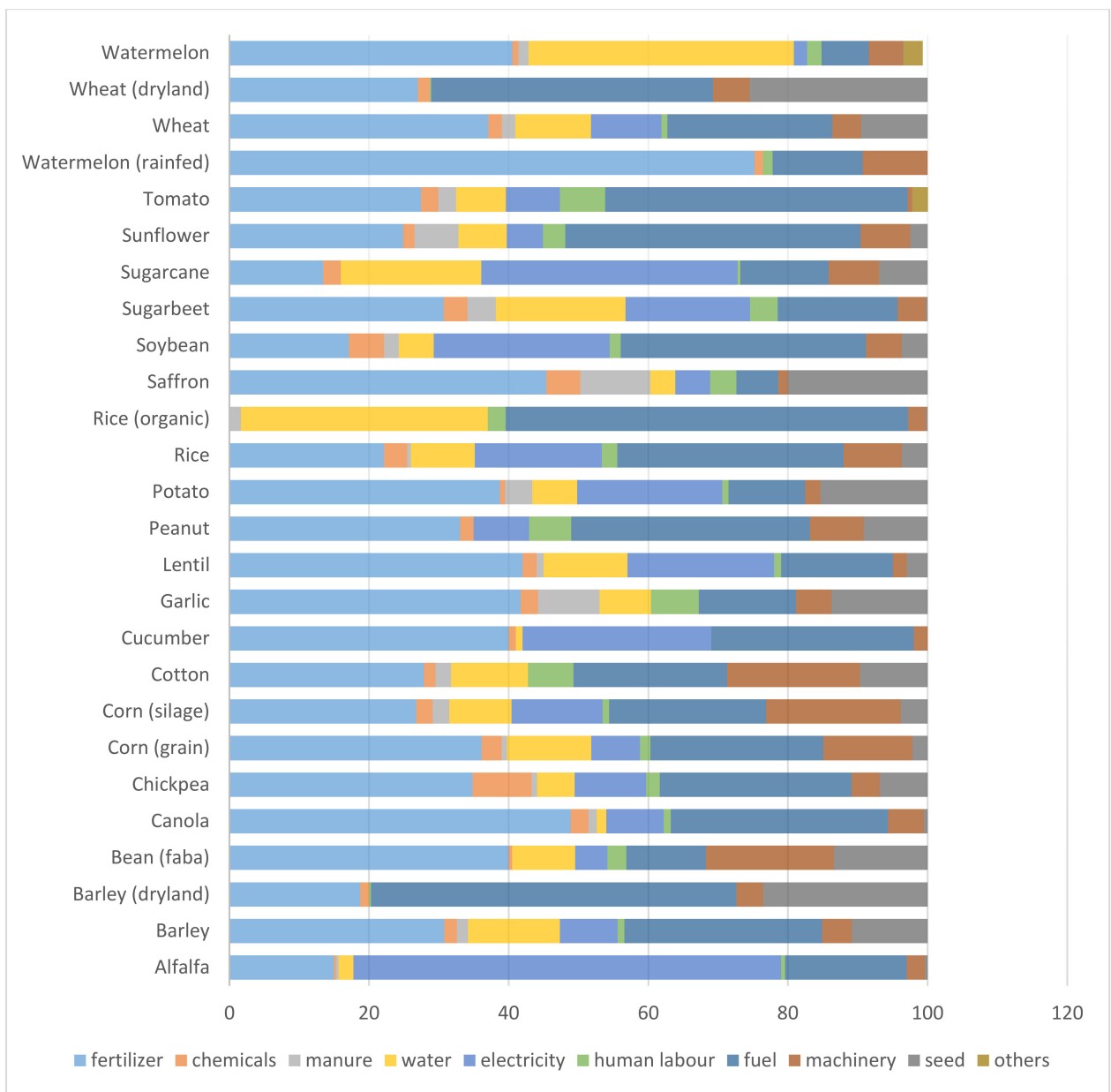

**Figure 4.** Share of inputs (%) in total energy input (TEI).

### 3.3.2. Environmental Impact

In addition to examining energy and economic indices, many of the articles we found addressed environmental issues of cropping systems. Here, we focus first on the 54 studies that investigated the environmental impacts associated with energy use. Production, transportation, storage, input distribution and machinery use consume fossil fuels and other energy sources that emit GHGs into the atmosphere. The most common method of calculating environmental impacts uses emission factors (Table 2) that are multiplied by the corresponding input inventory, such as diesel fuel, chemical fertilizer, biocides and irrigation water. The emissions are presented in kilograms of carbon equivalent for different farming activities and can be compared to alternatives, such as biofuels and renewable energy sources [26].

Among the reviewed papers, GHG emissions and global warming potential (GWP) are sometimes calculated manually using emission factors ($CO_2$ equivalents) or determined using a life cycle inventory (LCI) and software analysis. The latter allows a wide range of

environmental impacts to be assessed. Khoshnevisan et al. [27], for example, used Ecoinvent LCI and SimaPro Software for life cycle assessment (LCA) of rice production systems in northern Iran. Several authors have subsequently performed comprehensive studies of individual cropping systems with LCA [28,29]. Further, the energy and environmental impacts of different cropping systems have been compared using LCA [30,31].

Table 5 summarizes GHG emissions reported for crops studied in Iran. Researchers have applied emission factors to report GHG emissions of crops, including potato [26], cotton [32], wheat [33] and barley [34]. As indicated in Table 5, GHG emissions of some crops have been determined and reported in multiple studies. In such cases, the average is presented and is likely more reliable. It is notable that there is a wide range of $CO_2$ emissions reported across these crop production systems. For example, 181,190 and 558 kg $CO_2$-eq were produced per ha of alfalfa [35] and potato [36] production, respectively. In most cases, reported values for GHG emission in Iranian studies are similar to those of other studies around the world. For instance, Yousefi et al. [37] reported 2994.66 kg $CO_2$-eq per ha for corn production, while Camargo et al. [38] reported that the GHG emissions associated with corn production ranged from 2441 to 4201 kg $CO_2$-eq per ha per year in several publications. However, there are some papers with results differing significantly with those reported for GHG emissions for agricultural systems outside of Iran in crops like alfalfa and corn [39,40].

**Table 5.** The amounts of GHG emissions reported for farm crops of Iran.

| Crop | GHG Emission | | References |
|---|---|---|---|
| | kg $CO_2$ t$^{-1}$ (Mass Base) | kg $CO_2$ ha$^{-1}$ (Land Base) | |
| Alfalfa | 52.09 | 181,190 | [35,41] |
| Barley | - | 628 | [34,42] |
| Canola | - | 836.6 | [28,42,43] |
| Chickpea | 3032.6 | 6884.14 | [44] |
| Corn | - | 2994.66 | [37] |
| Cotton | - | 1195.25 | [32] |
| Lentil | 3593.2 | 7259.31 | [45] |
| Peanut | 311.19 | 697 | [25,46,47] |
| Potato | 116.4 | 558 | [26,36,48] |
| Rice | 1101 | 3197.00 | [27,42,49–51] |
| Saffron | - | 6545.8 | [52] |
| Soybean | 455,515 | 1197 | [42,53,54] |
| Sugar beet | - | 9847.77 | [55] |
| Sugarcane | - | 8249.12 | [56] |
| Tomato | 200 | - | [29] |
| Watermelon | - | 5299 | [6,7,9,11,12] |
| Wheat | 1600 | 2155 | [6,31,33,42,57–64] |
| Grape | 508.63 | - | [65] |
| Tobacco | 1883.90 | 3638.98 | [66] |

### 3.3.3. Data Envelopment Analysis

A summary of findings from DEA studies is presented in Figure 5, including estimated TE, PTE, and SE of crops, TES, and the share of each input in the total energy-saving ratio. As shown in the share of agricultural inputs in the total energy-saving ratio, electricity and fertilizer inputs have the most potential for energy reductions across the various crops. Potato has the highest potential to reduce energy consumption. By contrast, there is much less potential for peanut. TES in peanut farms of Guilan province were reported to be 9.15%, with machinery accounting for the largest fraction at 13.93% of the total, followed by biocides (10.23%) and chemical fertilizers (9.79%) [25].

Examples of complementary analysis with the DEA method include super-efficiency analysis by Mohammadi, Rafiee, Jafari, Dalgaard, Knudsen, Keyhani, Mousavi-Avval and Hermansen [55] and fractional regression model by Raheli et al. [67]. Mohammadi, Rafiee,

Jafari, Dalgaard, Knudsen, Keyhani, Mousavi-Avval and Hermansen [55] first applied LCA and DEA methodologies to soybean farming to benchmark the level of operational input efficiency and potential GHG emissions reductions by farmers. In addition, DEA has been widely used in combination with CO2-eq emissions data to determine the potential of GHG emissions reductions in agricultural systems [8,25,68].

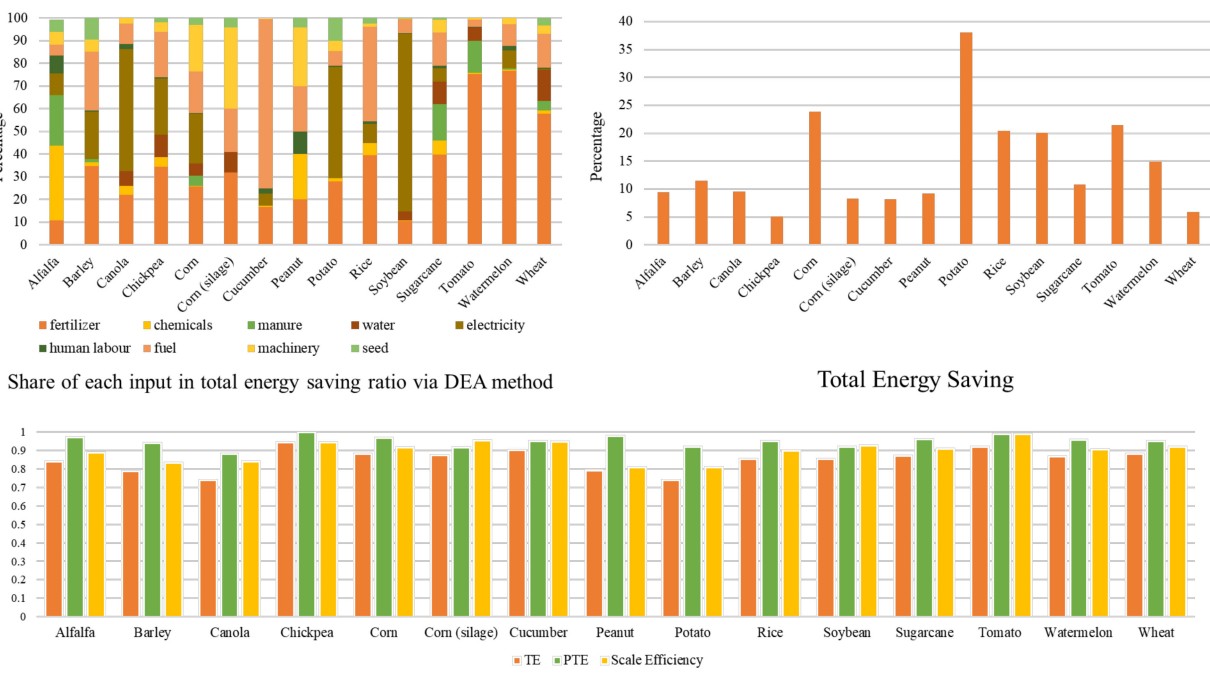

**Figure 5.** Summary of results from data envelopment analyses (DEA).

### 3.3.4. Economic Indicators

As the field of energy studies flourished in Iran, some researchers began adding economic indicators to their studies. Some studies reported costs of inputs in detail [69], while others only reported economic indicators. Various scales and units have been applied in reporting economic indices. For example, Ghorbani et al. [70] reported Total Production Cost, Gross Return and Net Return in $ ha$^{-1}$, $ kg$^{-1}$ and $ MJ$^{-1}$, respectively. Such economic analyses were common in studies from 2010 to 2012. Then, research interest gradually shifted to other techniques. Just two papers in 2013 and 2014 applied economic analysis [71,72], and none did in 2015. In 2016, Sahabi, Feizi and Karbasi [20] compared energy and economic aspects of saffron and wheat in northern Iran. In 2017, Mousavi-Avval et al. [73] used the same economic indicators to analyze canola production in northern Iran. In recent years, studies have become more varied, and new analyses have been added with environmental impacts highlighted.

### 3.3.5. GLM Results

In our review, we identified 21 different product types or crops among the articles to be used for modeling. We created eight linear regression models with the following energy and environmental indicators as dependent variables: GHG, SE, PTE, TE, NE, TEI, EP and EUE. The results of these models are presented in Table 6. We used three independent variables for regression modeling including region, year and crop. The "region" represents the climates of Iran, which includes 12 climates (Table 3). Four of the eight models predicted the dependent variable using the terms provided.

Of course, differences among regions in energy consumption and environmental emissions for the same product are not simply due to climate differences. There are other

differences, such as the level of technology used, planting methods, harvesting and storing, farmer knowledge, access to farming inputs, etc. However, because the articles reviewed did not always report the specific characteristics of cropping systems, we are limited in explaining the differences in energy and environmental indicators between different regions. The "year" refers to when the study was performed as reported in the article. The "crop" factor was influential in the regression models, which seems obvious because the amounts and types of inputs consumed and the production outputs of various products are fundamentally different from one to another. Thus, crop type would be expected to be a very important factor.

**Table 6.** General linear model results.

| Dependent Variables | | Source | | | | | | R Squared |
|---|---|---|---|---|---|---|---|---|
| | | Corrected Model | Intercept | Year [1] | Region | Crop | Region * Crop | |
| GHG | F | 5.668 ** | 2.233 | 2.256 | 9.003 ** | 5.433 ** | 8.257 ** | 0.886 |
| | Sig. | 0.000 | 0.155 | 0.153 | 0.000 | 0.001 | 0.003 | |
| SE | F | 3.143 | 3.882 | 3.865 | 0.751 | 3.995 | . | 0.962 |
| | Sig. | 0.268 | 0.188 | 0.188 | 0.571 | 0.216 | . | |
| PTE | F | 5.535 | 0.067 | 0.113 | 12.696 | 4.948 | . | 0.978 |
| | Sig. | 0.164 | 0.820 | 0.769 | 0.073 | 0.179 | . | |
| TE | F | 3.838 | 0.707 | 0.579 | 1.118 | 4.847 | . | 0.953 |
| | Sig. | 0.147 | 0.462 | 0.502 | 0.434 | 0.111 | . | |
| NE | F | 23.618 ** | 2.568 | 2.526 | 5.375 ** | 47.819 ** | 7.735 ** | 0.955 |
| | Sig. | 0.000 | 0.114 | 0.117 | 0.000 | 0.000 | 0.000 | |
| TEI | F | 3.997 ** | 0.064 | 0.073 | 1.56 | 5.355 ** | 1.556 | 0.766 |
| | Sig. | 0.000 | 0.801 | 0.787 | 0.137 | 0.000 | 0.071 | |
| EP | F | 0.284 | 0.394 | 0.396 | 0.329 | 0.237 | 0.232 | 0.201 |
| | Sig. | 1.000 | 0.532 | 0.531 | 0.97 | 1.000 | 1.000 | |
| EUE | F | 13.797 ** | 0.917 | 0.872 | 4.099 ** | 31.49 ** | 2.612 ** | 0.923 |
| | Sig. | 0.000 | 0.342 | 0.354 | 0.000 | 0.000 | 0.001 | |

** Significant at the 0.01 level (2-tailed). * Significant at the 0.05 level (2-tailed). [1] Year is covariate for all dependent variables.

In the model developed for GHG, the variables of region, crop and their interaction (crop*region) have significant effects. Therefore, there are significant differences between crops. Applications of different inputs, including chemicals and diesel fuel in field operations, can cause differences in GHG emissions. Agricultural activities inevitably result in emissions of all three greenhouse gases: carbon dioxide ($CO_2$), methane ($CH_4$) and nitrous oxide ($N_2O$). Methane is mainly released through farm manure; nitrous oxide is produced from the combustion of fossil fuels, manure, soil cultivation and decomposition of crop residue; and carbon dioxide is generated and released into the atmosphere through fossil fuels and the decomposition of crop residue.

The regression analysis indicates that GHG emissions differ across the regions of Iran, meaning that different climates can cause different GHG emissions. Climate is shaped by short- and long-term characters, including temperature, average rainfall, floods, drought, etc. All these climate characteristics impact food quality, distribution and access patterns [74]. Alboghdady and El-Hendawy Salah [75] used panel data from 20 countries in the Middle East and North Africa from 1961 to 2009 to assess the impact of climate change and variability on agricultural production. The results showed that a 1% increase in winter temperature results in a 1.12% decrease in agricultural production. Our analysis indicates that warmer and drier climates have more emissions. This may be due to the lack of optimal conditions for production and a resulting need to consume more inputs to compensate.

A more detailed comparison of wheat production across four climates is shown in Figures 6 and 7. In the semi-warm climate, where Isfahan province is located, five studies have been performed on GHG emissions from wheat production [33,58,59,76,77]. They

show that the consumption of livestock manure, chemical fertilizers and electricity is much higher than in other climates of Iran. It seems that the dry climate of Isfahan province increases the need to pump water from agricultural wells using electricity. Thus, emissions from electrical inputs are higher than in other climates.

On the other hand, Golestan province, which has a wet and mild climate, has the lowest electricity consumption. Data show that the amount of fuel consumed in Kermanshah province, located in a semi-arid climate, is much higher than other climates. Kermanshah farms also need irrigation from well sources, mainly due to drought. If the wells in this province are not electrified, higher fuel consumption results from pumping water. In Khuzestan province electricity consumption has not been reported, likely due to the abundance of surface water to irrigate farms.

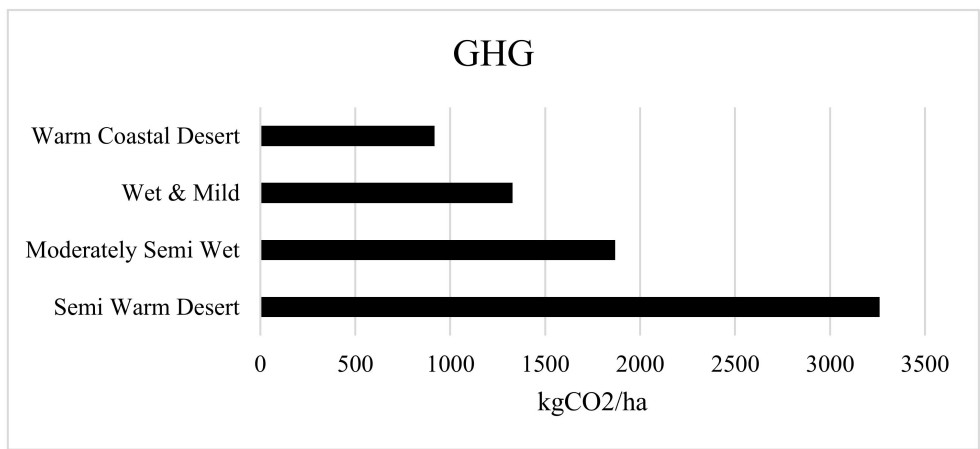

**Figure 6.** GHG emissions quantity for wheat in different climates.

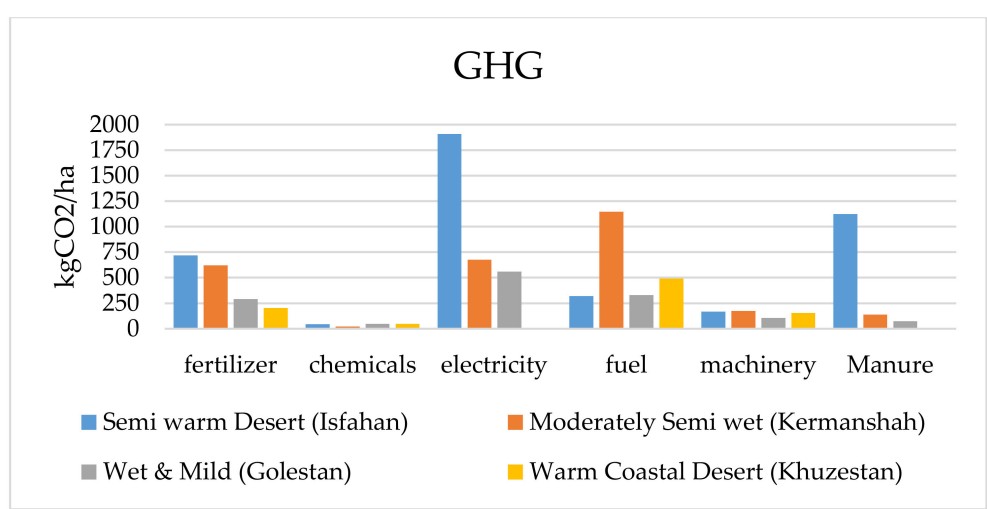

**Figure 7.** GHG quantity of inputs in different climates for wheat.

Examination of the NE regression model shows that, like the model for GHG emissions, the variables of region, crop and interactions of these two factors have significant effects. The same is true for the EUE model. However, for the input energy indicator, only the crop variable has a significant effect. Our four regression models show that climate factors, product type and which products are produced in which climate can predict GHG emissions, NE and EUE.

Our review revealed that from 2008 to 2018, ten studies were conducted on the energy consumption patterns of potatoes in Iran. Most of these studies took place in the main centers of potato production, namely Ardabil, Hamedan and Isfahan provinces. Comparing

the EUE in these areas indicates that Ardabil province has the best conditions (Figure 8). After Ardabil are Isfahan, Hamedan and Tehran provinces.

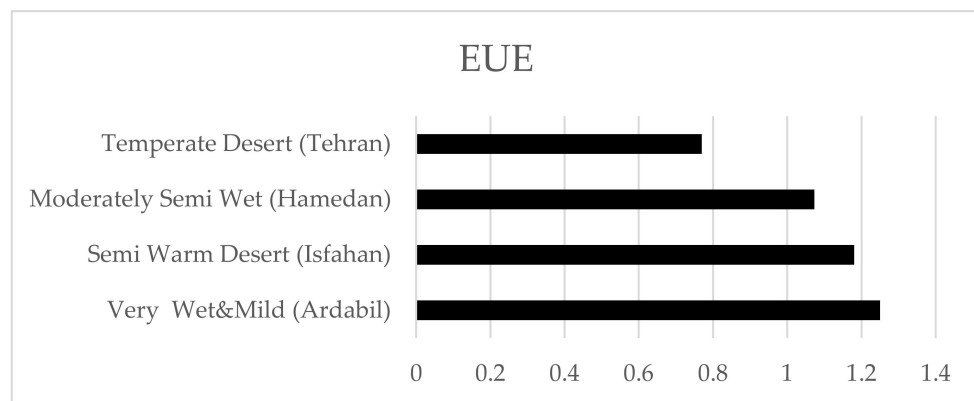

**Figure 8.** Energy use efficiency of potato production in different climates of Iran.

### 3.3.6. Production Function and Sensitivity Analysis

After calculating energy inputs, some researchers have sought to analyze relationships between energy inputs and crop output using the Cobb–Douglass production function [78,79]. The results of these studies are summarized in Table 7. Here, the highest t-ratio belongs to the most effective input influencing crop yield. A sensitivity analysis identifies how different values of an independent variable (energy input) can affect a specific dependent variable (crop yield). Sensitivity analyses were first used by Rafiee et al. [80] and Mobtaker, Keyhani, Mohammadi, Rafiee and Akram [2]. Mousavi-Avval et al. [81] reported the highest MPP index for fertilizer, water and machinery energy inputs on canola production. They showed that a 1 MJ overuse of each of these energy inputs will increase the canola production by 0.61, 0.24 and 0.93 kg, respectively. Generally, as can be seen from the results of sensitivity analyses, fuels, machines and chemical fertilizers had the greatest impact on crop yield.

**Table 7.** Highest t-ratio and MPP of crops produced in Iran.

| Crop | Highest t-Ratio | Highest MPP | References |
|------|-----------------|-------------|------------|
| Alfalfa | Diesel fuel | Diesel fuel | [82] |
| Alfalfa | Machinery | Machinery | [83] |
| Bean (red) | Machinery | Machinery | [84] |
| Canola | Nitrogen | - | [43] |
| Canola | Seed | - | [73] |
| Corn (silage) | Water | Seed | [40] |
| Corn (silage) | Biocide | Biocide | [3] |
| Cucumber | Chemicals and diesel fuel | Human labor | [85] |
| Cucumber | Machinery | - | [86] |
| Potato | Seed | - | [87] |
| Potato | Water | Seed | [88] |
| Potato | - | Fertilizer | [26] |
| Rice | Fuel | Fuel | [89] |
| Rice | Chemicals | - | [90] |
| Rice | Fertilizers | Fertilizers | [51] |
| Rice | Manure | - | [91] |

**Table 7.** *Cont.*

| Crop | Highest t-Ratio | Highest MPP | References |
|---|---|---|---|
| Soybean | Fertilizers | Machinery | [92] |
| Soybean | Seed | Seed | [93] |
| Soybean | Human labor | - | [54] |
| Sunflower | Diesel fuel | Chemicals | [21] |
| Watermelon | Human labor | Human labor | [11] |
| Watermelon | Chemicals | - | [12] |
| Wheat | Machinery | Machinery | [18] |
| Wheat | - | Fertilizers | [59] |
| Wheat (dryland) | - | Water | [94] |

*3.4. Risk of Bias across Studies*

Non-Parametric Test Results

Relationships between the three factors (region, year and crop) and each of the eight indicators (EUE, TEI, NE, EP, GHG, TE, PTE and SE) were tested using the Kruskal–Wallis test, the non-parametric equivalent of a one-way ANOVA. The results are summarized in Table 8. We can conclude that the regions have different NE, TEI and EUE. The crops also differ in terms of NE, TEI, EUE and EP (Figure 9). No significant relationship was found between the year variable and any of the energy and environmental indicators. This suggests that over the 10-year period, cropping systems were stable in terms of energy consumption patterns and environmental emissions. For comparison, Han and Wu [95] explored the impacts of changes in China's agricultural structure on factors such as energy intensity of agricultural production. Their results showed that the results of six vegetable production regions show great regional heterogeneity, which is mainly due to the scale economy effects and incremental increases in mechanization.

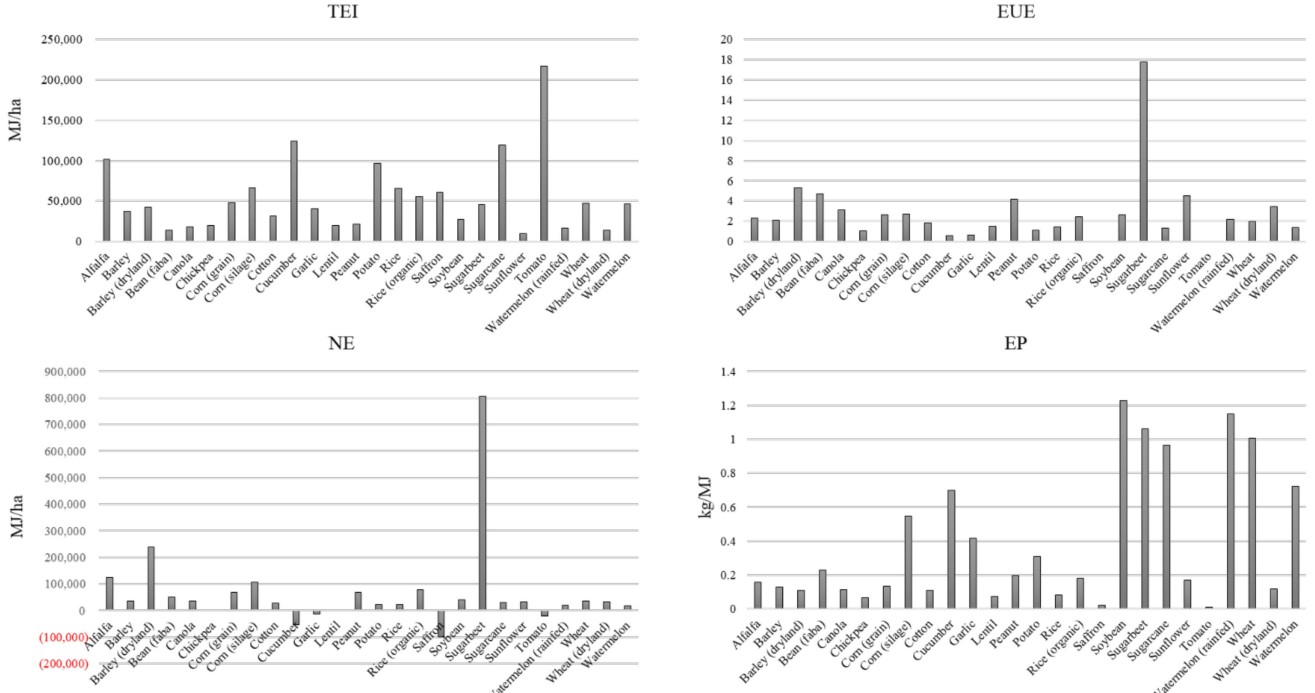

**Figure 9.** Average of energy indicators for different crops.

**Table 8.** Kruskal–Wallis test results.

| Test Variables | Grouping Variables | | | | | |
| --- | --- | --- | --- | --- | --- | --- |
| | Region | | Year | | Crop | |
| | Chi-Square | Asymp. Sig. | Chi-Square | Asymp. Sig. | Chi-Square | Asymp. Sig. |
| GHG | 7.254 | 0.403 | 3.595 | 0.731 | 16.340 | 0.231 |
| SE | 5.003 | 0.543 | 5.523 | 0.479 | 16.463 | 0.225 |
| PTE | 8.849 | 0.182 | 5.995 | 0.420 | 11.937 | 0.533 |
| TE | 2.322 | 0.888 | 3.107 | 0.795 | 13.216 | 0.431 |
| NE | 27.836 ** | 0.002 | 15.461 | 0.079 | 54.399 ** | 0.000 |
| TEI | 29.619 ** | 0.001 | 13.084 | 0.219 | 75.437 ** | 0.000 |
| EP | 13.292 | 0.208 | 7.453 | 0.682 | 59.404 ** | 0.000 |
| EUE | 30.082 ** | 0.001 | 11.793 | 0.299 | 61.791 ** | 0.000 |

** Significant at the 0.01 level (2-tailed).

## 4. Discussion

Agricultural systems are often described in terms of their diversity, or different types of systems, and their heterogeneity, or variation in the physical, biological, and human components within each type of system. The diversity and complexity of agricultural systems throughout the world can be tracked and evaluated with a variety of economic, environmental, and social performance indicators [96]. Our review revealed that in some published papers, values for energy and environmental indicators are reported, but no explanations are given for the causes or reasons for the status in these indicators such as Amanloo and Mobtaker [39,97,98]. In some cases, results have been compared with similar studies. For example, Mohammadi-Barsari, Firouzi and Aminpanah [12] stated that calculated energy productivity is higher for watermelon farms with advanced technology (0.66 kg MJ$^{-1}$) than for farms with less advanced technology (0.59 kg MJ$^{-1}$) in the semi-arid region of Hamadan province. Agricultural systems can vary in many aspects, such as farm size, seed type, date of cultivation, irrigation, tillage, harvest methods, etc., and some researchers have investigated these factors from the viewpoint of energy use, economics and other indicators, as shown in Table 9. Most of these studies occurred in the years 2011 and 2012 and often applied ANOVA and the Duncan multiple range test to compare means of classified groups. Statistical design has also been implemented to audit energy consumption and identify the share of energy inputs for different tillage systems [99].

**Table 9.** Energy indices in different cropping systems of Iran.

| Crop | Region | Classification | | IE (MJ ha$^{-1}$) | EUE | Reference |
| --- | --- | --- | --- | --- | --- | --- |
| **Potato** | Ardabil | Farm size | 0.1–2 ha | 39,677.9 | 2.72 | [13] |
| | | | 2.1–5 ha | 37,908.7 | 3.08 | |
| | | | 5 < ha | 37,482.5 | 3.62 | |
| **Red bean** | Kurdistan | Farm size | 0.1 | 105,540.2 | 0.18 | [84] |
| | | | 0.2 | 47,571 | 0.42 | |
| | | | 0.5 | 43,725.4 | 0.44 | |
| **Lentil** | Lorestan | Production method | Organic | 5062 | 2.12 | [100] |
| | | | Conventional | 6196.5 | 2.05 | |
| **Soybean** | Golestan | Irrigation system | Canal irrigation | 17,255.96 | 4.6 | [101] |
| | | | Pump irrigation | 38,266.71 | 2.15 | |

**Table 9.** *Cont*.

| Crop | Region | Classification | | IE (MJ ha$^{-1}$) | EUE | Reference |
|---|---|---|---|---|---|---|
| **Potato** | Hamedan | Technology level | High level | 153,071.4 | 1.14 | [87] |
| | | | Low level | 157,151.12 | 0.95 | |
| **Rice** | Mazandaran | Mechanization status | Traditional | 67,356.28 | 3 | [102] |
| | | | Semi-mechanized | 67,217.95 | 3.08 | |
| **Alfalfa** | Hamedan | Irrigation system | Traditional | 821,615.19 | 1.82 | [103] |
| | | | Modern | 723,254.38 | 2.06 | |
| **Wheat** | North Khorasan | Irrigation method | Irrigated | 45,367.63 | 1.44 | [70] |
| | | | Dryland | 9354.2 | 3.38 | |
| **Corn silage** | Alborz | Farm size | 5 > ha | 86,679 | 1.72 | [40,104] |
| | | | 5–10 ha | 65,845 | 2.29 | |
| | | | >10 ha | 54,499 | 2.8 | |
| **Rice** | Guilan | Seed type | *Hashemi* | 37,155.213 | 1.582 | [89] |
| | | | *Khazar* | 41,332.513 | 1.956 | |
| | | | *Hybrid* | 44,848.813 | 2.458 | |
| **Rice** | Guilan | Farm size | 0.5 > ha | 41,140 | 1.44 | [105] |
| | | | 0.5–1 ha | 40,433 | 1.47 | |
| | | | >1 ha | 36,428 | 1.69 | |
| **Corn** | Fars | Region | *Seyedan* | 41,631.97 | 2.6 | [106] |
| | | | *Houmeh* | 44,730.15 | 2.38 | |
| | | | *Pasargad* | 38,866.64 | 2.88 | |
| **Rice** | Mazandaran and Isfahan | Production method | Organic | 134,851.6 | 2.43 | [107] |
| | | | Conventional | 155,762.7 | 1.11 | |
| **Canola** | Khuzestan | Irrigation system | Irrigated | 28,944.65 | 1.28 | [108] |
| | | | Dryland | 18,557.72 | 0.81 | |
| **Potato** | Isfahan | Farm size | <1 ha | 51,460 | 1.3 | [26] |
| | | | 1–5 ha | 45,710 | 1.75 | |
| | | | >5 ha | 43,874 | 2.08 | |
| **Corn** | Alborz | Harvesting system | Combine harvesting | 49,303 | 5.15 | [104] |
| | | | Plot harvester | 49,448 | 4.4 | |
| | | | Two stage harvesting | 54,471 | 4.78 | |
| **Corn silage** | Tehran | Farm size | <10 ha | 38,841.5 | 3.11 | [109] |
| | | | 10–20 ha | 36,140.3 | 3.47 | |
| | | | 20–30 ha | 35,861.1 | 3.56 | |
| | | | >30 ha | 35,211.6 | 3.82 | |
| **Rice** | Mazandaran | Traditional production method | Average | 71,092.26 | 1.72 | [110] |
| | | | Native | 60,187.41 | 1.33 | |
| | | | High yield | 73,220.42 | 1.74 | |
| | | | Hybrid | 79,908.94 | 2.01 | |

**Table 9.** *Cont.*

| Crop | Region | Classification | | IE (MJ ha$^{-1}$) | EUE | Reference |
|------|--------|----------------|---|---------|-----|-----------|
| **Rice** | Mazandaran | Mechanized production method | Average | 79,460.33 | 1.63 | [110] |
| | | | Native | 69,181.23 | 1.26 | |
| | | | High yield | 82,005.42 | 1.63 | |
| | | | Hybrid | 87,186.06 | 1.94 | |
| **Wheat** | Isfahan | Farm size | Small | 80,400 | 0.38 | [57] |
| | | | Medium | 79,290 | 0.5 | |
| | | | Large | 81,110 | 0.56 | |
| **Rice** | Guilan | Land management | Traditional | 74,200 | 0.9 | [27] |
| | | | Consolidate | 57,000 | 1.6 | |
| **Rice** | Khuzestan | Planting method | Transplanting | 50,022 | 2.305 | [111] |
| | | | Direct seeding | 34,543 | 2.844 | |
| **Watermelon** | Khorasan and Semnan | Irrigation system | Full irrigation | 25,626 | 1.17 | [7] |
| | | | Reduced irrigation | 3129.3 | 4.08 | |
| **Soybean** | Golestan | Mechanization status | Modern mechanized | 29,532 | 1.53 | [54] |
| | | | Mechanized | 29,599 | 1.98 | |
| | | | Conventional (more tillage) | 15,369 | 3.03 | |
| | | | Conventional | 14,657 | 3.18 | |

Some researchers reported that larger-scale farms perform better according to energy indicators [40,105]. To investigate the effect of farm size and crop type on EUE, a regression model was developed using meta-data (Table 10). The regression model indicates that the size of the farm has no significant effect on energy efficiency, but the crop type does affect the EUE.

**Table 10.** General linear model result for energy use efficiency (EUE).

| Dependent Variables | | Corrected Model | Intercept | Crop Type | Farm Size | Crop Type Farm Size | R Squared |
|---------------------|---|-----------------|-----------|-----------|-----------|---------------------|-----------|
| EUE | F | 1.827 | 49.828 | 5.941 ** | 0.417 | 0.052 | 0.81 |
| | Sig. | 0.236 | 0.000 | 0.028 | 0.677 | 1.000 | |

** Significant at the 0.01 level (2-tailed).

For comparison, Ito [112] measured regional differences in agricultural productivity in China to test the validity of a hypothesis related to agricultural technology. Qiang et al. [113] compared the agricultural disasters in the north and south of China, and the results showed that the losses in the north increased by about 0.6% every ten years, close to twice that in the south of China. In addition, agriculture in northern China was more sensitive to precipitation change, while agriculture in southern China was more sensitive to temperature change. Other papers have investigated the effects of production parameters on the energy-use patterns. For example, Banaeian and Namdari [5] investigated how farming technology and ownership of machinery, tractors and land can affect energy-use patterns in watermelon. In analyzing the heterogeneity of agricultural technology, Fei and Lin [114] used meta-frontier DEA to measure agricultural energy efficiency. The results showed that energy efficiency in the eastern region of China was significantly higher than that of the western region. Based on the provincial panel data of 1995–2014, Diao et al. [115] analyzed the agricultural productivity and its regional differences in China.

Expanding cropland onto areas under natural ecosystems reduces carbon stocks in natural vegetation and soils, with the amount of carbon released and crop yields differing markedly between temperate regions and the tropics [116]. Cassman [117] indicates that precise management and improvements in soil quality are needed to achieve high yields without causing environmental damage. No-tillage has revolutionized agricultural systems because it allows individual producers to manage larger amounts of land with fewer inputs of energy, labor and machinery [118]. Lal [119] points out that not all conservation agriculture practices, and other resource conservation technologies are applicable across all farming systems. Sustainable land management practices such as crop diversification and chemical management are potentially efficient measures that lead to environmental and economic benefits [120]. In addition, synthetic N fertilizer is the major source of GHG emissions from crop production [121]. Better management in the use of chemical fertilizers on farms can be effective in improving productivity as well as protecting water resources and minimizing greenhouse gas emissions. Therefore, creating GHG profiles of different crops in Iran can greatly help in developing and improving national mitigation plans/policies that currently lack details pertaining to the agricultural sector.

In our analysis, as the results of the Kruskal–Wallis test showed, Iran's cropping systems have been almost constant over the 10-year period in terms of energy consumption patterns and environmental emissions based on the 21 crops evaluated. This apparent lack of improvement points to the need for researchers to identify methods to improve energy consumption patterns. Closer examinations of the characteristics of cropping systems should help researchers to identify and better understand energy consumption patterns. Differences in the indices for energy consumption and environmental emissions for similar products in different regions and years are not easily explained and will therefore require access to more precise information about the characteristics of agricultural systems.

As researchers continue in their efforts, they should pay attention to the following three points: (1) it is essential to understand the reasons for the current situation in order to devise reasonable, practical strategies for improvement; (2) farmers must be intimately involved in diagnosis of the problems and in devising improvement strategies; and (3) agricultural technologies and policies (and support systems) are complementary means of improving agricultural productivity and sustainability. If researchers can implement solutions to improve patterns of energy consumption on sample farms, they can serve as models for farmers. Agricultural systems, like any other system, need continuous analysis and improvement, so researchers must identify inputs with non-optimal consumption, design solutions to address the issues, and implement pilot projects to demonstrate improved performance.

Jones et al. [122] studied and summarized the history of agricultural systems modeling and identified lessons learned that can help guide the design and development of tools and methods for the next generation of agricultural systems. They emphasized that there are two broad categories that motivate agricultural model development: scientific understanding and decision/policy support. The complex and interacting dimensions of modeling agricultural systems are illustrated in Figure 10.

Agricultural systems are affected by many external drivers, which can be divided into four groups: technical, economic, environmental and social. These systems can also be classified and studied at various scales, such as field, farm, regional, national and global. Users include farmers, suppliers, agencies, governments and international companies, and they pursue different goals. At the field level, input consumption management is key, while at the farm level it is production optimization and business management. Shifting to the regional level, natural resource conservation, economic planning, and environmental and landscape management become the focus. At the national level, the priorities become trade policies, poverty alleviation and strategic planning, and globally, they are global business, climate change adaptation and food security to reliably feed more than 9 billion people. To improve the performance of agricultural systems, modeling and analysis at different scales

is necessary. The results of our review show that studies conducted in Iranian agricultural systems have been focused on field and farm scales.

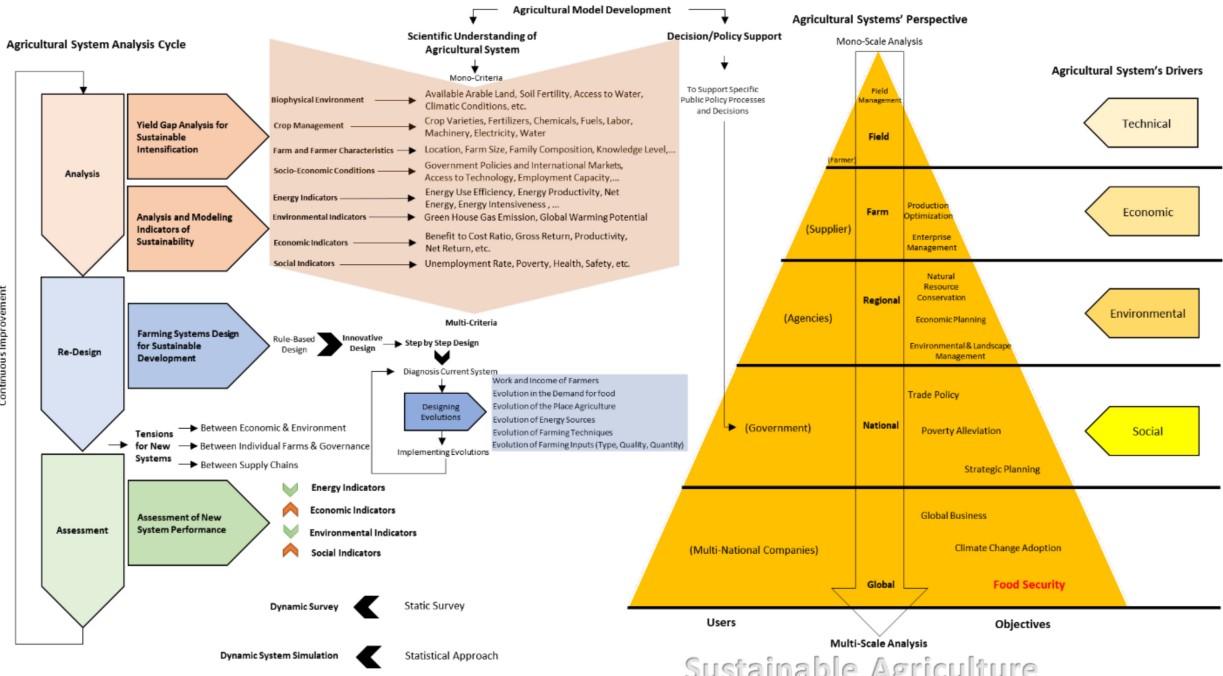

**Figure 10.** Dimensions and requirements of modeling and analysis of agricultural systems.

Analysis and modeling of agricultural systems should exist within a continuous cycle of at least three main phases: system analysis, system redesign and system assessment. System analysis consists of two steps: (1) yield gap analysis for sustainable intensification and (2) analysis of and modeling indicators of sustainability. Since many different factors affect these systems, it is necessary to shift the analysis and modeling from a single criterion to multiple criteria. At the field or farm level, biophysical models are used to analyze responses similar to the way that experiments on the real systems would be analyzed. Biophysical environment indicators include available arable land, soil fertility, access to water, climatic conditions, etc. Crop management indicators are also used at field and farm scales and consist of: crop varieties, fertilizers, chemicals, fuels, labor, machinery, electricity and water. Indicator categories for farm and farmer characteristics include location, farm size, family composition, knowledge level, etc. Socio-economic indicators that have not generally been considered in the articles reviewed but have important effects on output are government policies and international markets, access to technology, employment capacity, etc. Other indicators mainly used to model sustainability include the following energy, environmental, economic and social indicators: EUE, EP, NE, energy intensiveness, GHG emissions, GWP, benefit-to-cost ratio, gross return, productivity, net return, unemployment rate, poverty, health and safety.

Jones, Antle, Basso, Boote, Conant, Foster, Godfray, Herrero, Howitt, Janssen, Keating, Munoz-Carpena, Porter, Rosenzweig and Wheeler [122] demonstrated that a minimum set of components is needed to develop agricultural-system models that are common across various applications. These include crop models that combine weather, soil, genetic and management components to simulate yield, resource use and outputs of nutrients and chemicals to surrounding water, air and ecological systems. These crop models need to account for weed, pest and disease pressures and predict performance resulting from a range of inputs and practices that represent subsistence to highly controlled, intensive production practices.

The second phase of the analysis cycle is redesigning the system. Redesign of farming systems must: (1) prepare a diversity of solutions for different futures and leave the choices to farmers and other stakeholders and (2) support farmers and other stakeholders in building their own systems, adapting them to their own situation, and deciding on their own compromises as they rely on their knowledge and experiences along with scientific knowledge [123].

Two methods for redesigning agricultural systems have been proposed: rule-based design and innovative design. Most attention has been paid to the latter. A step-by-step approach organized in a progressive transition towards more innovative systems is best [123]. System redesign begins with diagnosis followed by a determination of what steps in the system are challenging and what technical or managerial measures are needed to address them. Accordingly, evolutionary measures are designed and implemented. Then, new diagnoses are performed, and the redesign continues in the same way. These actions benefit from progress made in recent years in agricultural systemic analysis methods in situ, which make it easier to carry out precise and reliable diagnoses.

The third phase of the agricultural-systems analysis cycle is to evaluate the performance of the new system. In general, the new system is evaluated in terms of the same energy, economic, environmental and social indicators. After ensuring the desired effects of the redesign measures, the whole system is analyzed again. This cycle will continuously improve the system and move towards a more sustainable agricultural system.

### 4.1. Summary of Evidence

Energy Indicators Status

Our analysis revealed that tomato is the most demanding energy consumer per hectare among the crops of Iran, and sugarcane, cucumber and alfalfa follow. By contrast, sunflower is the least energy-demanding crop in Iran. The overall average TEI in farm crops of Iran during the study period was 48,029 MJ ha$^{-1}$. Sugar beet has the highest energy-use efficiency due to its output energy equivalent of 16.80 MJ kg$^{-1}$ [124]. Some crops have a negative NE, including cucumber, garlic, saffron,] and tomato. The lowest EUE belongs to saffron, due to traditional production methods in Iran, in which most operations other than land preparation and fertilizer spraying are performed by human labor [52]. Further, saffron yields only about 3.7 kg ha$^{-1}$ under the best conditions. Although the cultivation of saffron is not efficient from the viewpoint of energy balance, it is significant economically. Khanali, Movahedi, Yousefi, Jahangiri and Khoshnevisan [52] stated that to create a better balance between the energy inputs and saffron yield, efforts should focus on increasing saffron yield and subsequently its energy productivity.

The heterogeneity in energy and environmental emissions indicators for the same product in different regions and different years is not easy to explain. Identifying reasons for these differences requires access to more information about the characteristics of agricultural systems and ecosystems. Since the studies performed in different products have usually been in different climates, the average values calculated in this article for the eight indicators can only be used in general.

### 4.2. Limitations

Calculating energy indicators using local rather than national or international energy equivalents would be more precise. There is a critical need to estimate local energy equivalents for inputs in each country, including Iran. Thus, energy conversion factors for food, not just agricultural products, should be developed and used at a wide range of geographic scales, from local to regional to national levels [125].

Systems analysis requires data collection that reflects a specific system's actual behavior. However, due to the vastness and complexity of agricultural systems on the ground and their geographical dispersion, collecting data from agricultural systems is challenging and costly. To overcome this difficulty, some researchers have identified innovative data collection approaches, including crowd-sourcing and remote- and close-sensing [126]. Such

innovative approaches are particularly relevant and promising for collecting data on the variability in crop management among different farms and farmers.

## 5. Conclusions

In this review, we examined all published papers on crop energy dynamics in Iran over a decade (2008–2018). Although many papers were published on the topic, there had been no systematic review that comprehensively collected and analyzed all results to suggest directions for future research. Energy production and consumption is costly and leads to negative environmental impacts; thus, researchers have investigated economic and environmental aspects alongside energy. In this paper, we summarized the shares of different energies across different products and regions and used DEA as an approach to find opportunities to improve the energy consumption patterns. Comparison of results of reviewed papers showed that the tomato is the most demanding energy consumer per hectare among the crops of Iran, and sugarcane, cucumber and alfalfa follow. By contrast, sunflower is the least energy-demanding crop in Iran. Our review revealed that in some published papers, values for energy and environmental indicators are reported but no explanations are given for the causes or reasons for the status in these indicators. Closer examinations of the characteristics of cropping systems should help researchers to identify and better understand energy consumption patterns. Differences in the indices for energy consumption and environmental emissions for similar products in different regions and years are not easily explained and will therefore require access to more precise information about the characteristics of agricultural systems. The meta-analysis section identifies patterns in the results of the studies included, sources of disagreement among those results, and other interesting relationships. Reasons for observed heterogeneity among the results of similar studies were interpreted using the nonparametric Kruskal–Wallis test and meta-regression. We created eight linear regression models with GHG, SE, PTE, TE, NE, TEI, EP and EUE. For meta-analysis, the data were analyzed using three factors: crop (product), year and regional climate. We can conclude that the regions have different NE, TEI and EUE. The crops also differ in terms of NE, TEI, EUE, as well as EP. No significant relationship was found between the year variable and any of the energy and environmental indicators. This result showed that over the 10-year period, cropping systems were stable in terms of energy consumption patterns and environmental emissions.

The overarching general lessons from our comprehensive review are as follows: (1) cropping systems in iran are very diverse; (2) cropping systems are affected by a wide variety of drivers; (3) modeling and analysis of cropping systems should give comprehensive attention to factors affecting the system, and it is essential to understand the reasons for the current situation in order to devise reasonable, practical strategies for improvement; (5) cropping systems can be modeled and analyzed at different levels from field to global scales and farmers must be intimately involved in diagnosis of the problems and in devising improvement strategies; (6) analyzing cropping systems at multiple scales is recommended; (7) indicators of cropping systems can be classified into four groups, which are energy, environmental, economic and social; (8) all four groups of indicators should be used to analyze and understand cropping systems; (9) systems analysis should be done cyclically and in a step-by-step process to design corrective actions and solve problems; (10) agricultural technologies and policies (and support systems) are complementary means of improving agricultural productivity and sustainability

**Author Contributions:** Conceptualization, M.Z.; methodology, M.Z.; validation, S.C.; analysis, M.Z. and N.B.; investigation, N.B.; resources, N.B.; data curation, N.B.; writing—original draft preparation, M.Z.; writing—review and editing, M.Z. and S.C.; visualization, M.Z. All authors have read and agreed to the published version of the manuscript.

**Funding:** This research received no external funding.

**Informed Consent Statement:** Not applicable.

**Conflicts of Interest:** The authors declare no conflict of interest.

**Abbreviations**

| | |
|---|---|
| IE | Input Energy |
| EUE | Energy Use Efficiency |
| NE | Net Energy |
| FYM | Farm Yard Manure |
| IDE | In-Direct Energy |
| NRE | Non-Renewable Energy |
| ANOVA | Analysis of Variance |
| TE | Technical Efficiency |
| SE | Scale Efficiency |
| GHG | Greenhouse Gas |
| DE | Direct Energy |
| RE | Renewable Energy |
| TEI | Total Energy Input |
| PTE | Pure Technical Efficiency |
| EP | Energy Productivity |
| MPP | Marginal Physical Productivity |
| DEA | Data Envelopment Analysis |
| GLM | General Linear Model |
| CSA | Climate Smart Agriculture |
| LCA | Life Cycle Assessment |
| TES | Total Energy Savings |
| PRISMA | Preferred Reporting Items for Systematic Reviews and Meta-Analyses |

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
