# Peer review of "Meta-Analysis on Energy-Use Patterns of Cropping Systems in Iran"

_sustainability, doi:10.3390/su13073868_

Round 1

Reviewer 1 Report

The submitted study is interesting and original. Currently, the study about a meta-analysis of energy-consumption and environmental-emissions issues is needed.
Nevertheless, there are some improvements required before publication, as follows:
i) the issue considered in the article is not embedded in the literature on the subject. The introduction does not relate the study to the literature on the subject and does not indicate the significance of the issue in respect of modern methodological and theoretical concepts. The Author(s) should provide in-depth explanation why these studies are needed?
ii) The conclusion requires a more detailed explanation. I would recommend to highlight empirical research results.
iii) English language and style are minor spell check required.

Author Response

Thanks a lot for valuable comments. Answers are provided in BLUE and changes in the manuscript is highlighted by “Track Changes” option of MS-Word.

Reviewer 1

The submitted study is interesting and original. Currently, the study about a meta-analysis of energy-consumption and environmental-emissions issues is needed.
Nevertheless, there are some improvements required before publication, as follows:
i) the issue considered in the article is not embedded in the literature on the subject. The introduction does not relate the study to the literature on the subject and does not indicate the significance of the issue in respect of modern methodological and theoretical concepts. The Author(s) should provide in-depth explanation why these studies are needed?

Thanks a lot for your comment. We described the necessity of our study in the Introduction by adding following paragraph:

In recent years the number of publications on energy and environmental aspects of agricultural systems has increased in Iran, particularly on energy consumption patterns and environmental emissions [1]. The considerable number of publications in this field of research has necessitated a comprehensive and analytical study of the various dimensions of such studies. The method of conducting studies in this field has evolved in recent years. However, there are challenges facing researchers in this field. The first question of the present study is that despite the wide range of indicators and methods used by previous researchers, have all aspects of cropping systems been examined or are there still issues in the systems analysis process that have been neglected? Given the great variety of production systems in the agricultural sector, have the necessary standards been met by researchers in sampling and data collection? In terms of the system analysis process, it is incomplete to study the current state of a system without intervening to improve a system. Therefore, the next question is whether in the studies, interventions in the energy consumption pattern have been done by researchers and the effect of those interventions on improving the energy consumption pattern, economic model and reducing environmental emissions has been studied? Another important question is whether the hypothesis has been made about the reason for the formation of energy consumption patterns in the products under study? Are solutions proposed to improve the current state of a system in the studies? Have social indicators been studied in studies in this field? Has the selection of the region and the product under study been based on regional and national needs and necessities? In response to the above questions, and in order to better directions for future research and prevent duplicate studies that have sometimes been observed in this area, it is necessary to provide researchers with sufficient information about the background of the studies. Therefore, using a systematic method, the present article provides readers with a comprehensive picture of the research background and energy consumption status in Iranian cropping systems. The sections that have been neglected in previous research are also mentioned in this article, which can be used as a basis for future research by researchers to improve the quality of studies in this field. A new procedure to supplement the shortcomings of previous studies is introduced in this article.

  1. ii) The conclusion requires a more detailed explanation. I would recommend to highlight empirical research results.

Ok, we added the following paragraph to the conclusion:

Comparison of results of reviewed papers showed that tomato is the most demanding energy consumer per hectare among the crops of Iran and sugarcane, cucumber and alfalfa follow. By contrast, sunflower is the least energy-demanding crop in Iran. Our review revealed that in some published papers, values for energy and environmental indicators are reported but no explanations are given for the causes or reasons for the status in these indicators. Closer examinations of the characteristics of cropping systems should help researchers to identify and better understand energy consumption patterns. Differences in the indices for energy consumption and environmental emissions for similar products in different regions and years are not easily explained and will therefore require access to more precise information about the characteristics of agricultural systems. The meta-analysis section identifies patterns in the results of the studies included, sources of disagreement among those results, and other interesting relationships. Reasons for observed heterogeneity among the results of similar studies was interpreted using the nonparametric Kruskal-Wallis test and meta-regression. We created eight linear regression models with GHG, SE, PTE, TE, NE, TEI, EP and EUE. For meta-analysis, the data were analyzed using three factors: crop (product), year and regional climate. We can conclude that the regions have different NE, TEI and EUE. The crops also differ in terms of NE, TEI, EUE, as well as EP. No significant relationship was found between the year variable and any of the energy and environmental indicators. This result showed that over the 10-year period, cropping systems were stable in terms of energy consumption patterns and environmental emissions.

iii) English language and style are minor spell check required.

Ok, done.

Reviewer 2 Report

General comment

The paper is meta-analysis of energy-consumption and environmental-emissions based on data collected from articles published in ten years. The text and the structure of the manuscript is clear and easily readable. In my opinion the article is suitable for the publication on "Sustainability".

The authors have correctly faced the difficulties caused by the great heterogeneity of the available data, applying some statistical techniques necessary for the purpose.

Specific comments

  • Line 78. Please, add a bibliographic reference for Data Envelopment Analysis (DEA).
  • Line 180 and following. It is not clear to me why the Authors has separated the co-occurrence terms analysis between keywords and title/abstract. Is there a reason for this approach?
  • Table 9. I was surprised by a so high value of Input Energy for alfa-alfa. Is there a specific comment for this?

Author Response

Thanks a lot for valuable comments. Answers are provided in BLUE and changes in the manuscript is highlighted by “Track Changes” option of MS-Word.

Reviewer 2

The paper is meta-analysis of energy-consumption and environmental-emissions based on data collected from articles published in ten years. The text and the structure of the manuscript is clear and easily readable. In my opinion the article is suitable for the publication on "Sustainability".

The authors have correctly faced the difficulties caused by the great heterogeneity of the available data, applying some statistical techniques necessary for the purpose.

Specific comments

  • Line 78. Please, add a bibliographic reference for Data Envelopment Analysis (DEA).

Ok, The DEA is described well in following reference which cited in line 78:

  1. Banaeian, M. Omid and H. Ahmadi. Greenhouse strawberry production in Iran, efficient or inefficient in energy. Energy Efficiency 2012 Vol. 5 Issue 2 Pages 201-209.

  • Line 180 and following. It is not clear to me why the Authors has separated the co-occurrence terms analysis between keywords and title/abstract. Is there a reason for this approach?

To cover all the key words in this branch of science, all the sections that could be searched by the VosViewer software, the abstract, title and keywords were analyzed. Both analyses, keywords and title/abstract co-occurrence analysis, were performed to find the structure of the relationship between commonly used words in this field of research. The results of this analysis were used to better search articles in databases and review research literatures.

Co-occurrence analysis of authors was also performed to find the most important researchers in this field in Iran. In this way, researchers can select suitable collaborators from the most prominent researchers in this field for their future research.

  • Table 9. I was surprised by a so high value of Input Energy for alfa-alfa. Is there a specific comment for this?

The value of Input Energy for alfalfa is too high, because they calculated it for seven years’ period. You can see they stated this point in the abstract of their paper:

“Total energy used in whole production life of alfalfa was 821615.19 MJ/ha in Group I and 723254.38 MJ/ha in Group II.”

Reference: H. ghasemi Mobtaker, A. Akram, A. Keyhani and A. Mohammadi. Energy consumption in alfalfa production: A comparison between two irrigation systems in Iran. African Journal of Agricultural Research 2011 Vol. 5 Issue 1 Pages 47-51.
